# Efficacy and Safety of Epidermal Growth Factor Receptor (EGFR)-Tyrosine Kinase Inhibitor Combination Therapy as First-Line Treatment for Patients with Advanced *EGFR*-Mutated, Non-Small Cell Lung Cancer: A Systematic Review and Bayesian Network Meta-Analysis

**DOI:** 10.3390/cancers14194894

**Published:** 2022-10-06

**Authors:** Jianchao Xue, Bowen Li, Yadong Wang, Zhicheng Huang, Xinyu Liu, Chao Guo, Zhibo Zheng, Naixin Liang, Xiuning Le, Shanqing Li

**Affiliations:** 1Department of Thoracic Surgery, Peking Union Medical College Hospital, Chinese Academy of Medical Sciences and Peking Union Medical College, Beijing 100730, China; 2Chinese Academy of Medical Sciences and Peking Union Medical College, Beijing 100005, China; 3Department of Thoracic and Head and Neck Medical Oncology, The University of Texas MD Anderson Cancer Center, Houston, TX 77030, USA

**Keywords:** EGFR-TKI, combination therapy, Bayesian network meta-analysis, non-small cell lung cancer

## Abstract

**Simple Summary:**

As clinical practice lacks direct data for head-to-head comparisons across targeted combination approaches, we performed a Bayesian network meta-analysis to indirectly analyze the efficacy and safety of various combination therapies. A total of 12 modes of combination TKI therapy are compared. We find that radiation combined with EGFR-Tyrosine Kinase Inhibitor (TKI), is superior to other combination treatments in progression free survival (PFS) and overall survival (OS). TKI combined with chemotherapy may improve objective response rate (ORR), PFS, and OS, but may increase grade 3 or higher hematological side events. Combining efficacy and safety, we show that macro-monoclonal antibody antiangiogenic drugs, such as bevacizumab or ramucirumab, may be the ideal combination treatment choice, while the efficacy of small molecule inhibitors, such as apatinib, needs more investigation. Our findings have significant implications for the clinical management of non-small cell lung cancer patients, providing new ideas and evidence for doctors’ decision making, as well as pointing to future research options.

**Abstract:**

(1) Background: Several randomized controlled trials (RCTs) have been conducted in combination with Efficacy and Safety of Epidermal Growth Factor Receptor(EGFR)-Tyrosine Kinase Inhibitor (TKI) for the first-line treatment of patients with advanced non-small cell lung cancer; however, head-to-head comparisons of combination therapies are still lacking. Therefore, this study aims to compare the efficacy and safety of various combination treatments. (2) Methods: We conducted a systematic review and Bayesian network meta-analysis by searching MEDLINE, EMBASE, and COCHRANE for relevant RCTs. (3) Results: TKI combined with antiangiogenic therapy, chemotherapy, or radiation achieved a significant benefit compared with TKI alone for progression free survival (PFS). A combination with radiation yielded better benefits in PFS than any other treatment. In terms of overall survival (OS), only the combination with pemetrexed and carboplatin (HR = 0.63, 95% credible interval 0.43–0.86)/radiation (0.44, 0.23–0.83) was superior to TKI alone. All of the combination therapies may increase the incidence of ≥Grade 3 AEs, as the pooled RRs are over 1; different toxicity spectrums were revealed for individual treatments. (4) Conclusions: The TKI combination of radiation/pemetrexed and carboplatin could provide the best antitumor effects among the first generation TKI-based treatments. Considering safety, ramucirumab and bevacizumab may be the ideal additions to TKIs (systematic review registration: PROSPERO CRD42022350474).

## 1. Introduction

Lung cancer is the leading cause of cancer morbidity and mortality among men, and the second for mortality among women [1], and non-small cell lung cancer (NSCLC) accounts for approximately 85% of overall reported lung cancer cases. In 2009, the Iressa Pan-Asian Study (IPASS) established a new standard of care for the first-line treatment of patients with NSCLC who harbor activating epidermal growth factor receptor (*EGFR*) mutations, which are primarily observed in subjects who are Asian, women, former light or non-smokers, and adenocarcinoma patients [2,3]. Over the past decades, the efficacy of target therapy has been gradually demonstrated to be superior to standard traditional chemotherapy in several other independent, international, randomized controlled clinical trials, such as OPTIMAL, CONVINCE, and LUX-lung 3/6 studies [4,5,6]. Meanwhile, the treatment-related adverse events (trAEs) are far fewer than those from chemotherapy. However, the inevitability of acquired drug resistance to EGFR-tyrosine kinase inhibitors (EGFR-TKIs) remains a severe challenge. The duration of EGFR-TKI therapy has been shown to correlate positively with a better prognosis or long-term survival of patients [7,8]. Consequently, synergistic combinations of EGFR-TKIs with other treatments involving different modes of action, including chemotherapy, monoclonal antibodies, radiotherapy, and specific pathway inhibitors, have been studied as first-line possibilities for overcoming drug resistance and extending survival [9].

More than a dozen randomized controlled trials (RCTs) of EGFR-TKI in combination with other treatment have been conducted to compare the benefits of efficacy and safety of combination therapy over EGFR-TKI monotherapy. However, head-to-head comparisons between these combinations are still lacking, and there is no high-level direct comparative evidence-based medical proof to support these benefits. Furthermore, multiple studies have shown that the efficacy of different regimens varies with the diverse kinds of patients in terms of gender, smoking status, brain metastasis status, and various *EGFR* mutation statuses, including two classic types (19 deletion and 21 Leu858Arg mutations).

Although a few previous similar meta-analyses analyzed the benefits and disadvantages of first-line treatment for *EGFR*-mutated advanced NSCLC through indirect comparisons, there were still a large number of important RCTs whose results had not yet been published and therefore, required immediate updating [10,11]. Thus, we conducted this network meta-analysis [12], which is widely used in the absence of data from head-to-head trials, integrating the most recent results from RCTs and synthesizing indirect evidence to investigate the efficacy and safety of combination therapies of EGFR-TKI as a first-line treatment in patients with advanced *EGFR* mutated NSCLC, drawing more robust conclusions for determining the best clinical choice. Then a subgroup analysis by gender, smoking status, brain metastasis status, and 19 deletion/21 Leu858Arg mutations was conducted to find a logical conclusion.

## 2. Materials and Methods

This network meta-analysis was reported following the PRISMA 2020 (preferred reporting items for systematic reviews and meta-analyses 2020) checklist extension statement for network meta-analysis (Appendix A) [13]. It has been registered on PROSPERO with identification number CRD42022350474. The protocol of this meta-analysis is available at https://www.crd.york.ac.uk/prospero/display_record.php?ID=CRD42022350474.

### 2.1. Retrieval Strategy

We retrieved research from Medline (Ovid), Embase, the Cochrane Library, and ClinicalTrials.gov database up to 3 July 2022 in all languages. The reference lists of included studies were also examined for additional articles. As per the guideline of PICOS, the main keywords are “non-small cell lung cancer”, “*EGFR*”, “TKI”, “gefitinib”, “erlotinib”, “icotinib”, “afatinib”, “dacomitinib”, “osimertinib”, and their derivatives. The details of the retrieval strategy are available in Appendix A.

### 2.2. Inclusion and Exclusion Criteria

Studies meeting all the following inclusion criteria were included:

(1) RCTs that enrolled patients with histological or cytological confirmed EGFR-mutated advanced (stage III/IV/recurrent) NSCLC;

(2) Studies containing two or more arms of first-line treatment which involved the TKI therapy (except for research not concerning the combination therapies);

(3) Trials that reported at least one of the following outcomes:

A. Progression free survival (PFS), defined as the time from the date of randomization to the first date of disease progression or death from any cause.

B. Overall survival (OS), defined as the time from randomization to death from any cause.

C. Objective response rate (ORR), defined as the rate of patients who achieved the complete or partial response according to RECIST 1.1 (response evaluation criteria in solid tumors 1.1).

D. Adverse events of grade 3 or higher (≥Grade 3 AEs) based on the National Cancer Institute Common Terminology Criteria for adverse events, version 4.0.

Some additional exclusion criteria were also formulated: 

(1) Studies terminated due to failure of enrollment;

(2) Studies whose results were only reported on meetings briefly, with further information unavailable;

(3) Studies that mixed different generation TKIs in one treatment arm;

(4) Studies in which *EGFR*-mutated patients were subgroup populations.

(5) Studies exploring the sequence in combination therapy.

### 2.3. Data Extraction 

Two authors (Li and Xue) independently viewed the retrieved results to sift out the proper articles and extracted data from each one. Discrepancies were resolved by discussion with each other. 

The collected data from each study included the first author, year of publication, region, study design, some baseline messages, PFS, OS, ORR, ≥Grade 3 AEs incidence, interruption rate of TKI owing to AEs, and the incidence of developing the T790M mutation in patients with the first/second generation TKIs resistance. PFS was the primary outcome. Hazard ratio (HR) and its 95% confidence interval (95% CI) for PFS and OS, and relative risk (RR) for other outcomes presented in binary data, were used for analysis. We chose relative risk (RR) as the measure of effect because the odds ratio (OR) could drastically exaggerate the results, leading to a false positive conclusion, especially for the outcomes with a high incidence, such as ORR [14]. Subgroup information was also collected. The missing information was searched in the meetings’ abstracts, such as the American Society of Clinical Oncology (ASCO), the European Society of Medical Oncology (ESMO), and the World Conference on Lung Cancer (WCLC) or registration website; data still unavailable was skipped in our analysis. Data with longer follow-up times was used, if the research had several published reports.

CTONG0901, ICOGEN, and WJOG5108L trials indicated that the first generation of EGFR-TKIs, including gefitinib, erlotinib, and erlotinib, showed no statistically significant changes in PFS and OS in patients with *EGFR*-positive NSCLC [15,16,17]. To facilitate later analysis, we regarded the first generation of EGFR-TKIs in all included studies as a single arm.

### 2.4. Quality Assessment

Two authors (Li and Xue) independently assessed each included article using the Risk of Bias tool 2 (RoB 2) provided by Cochrane Collaboration (Appendix A) [18]. Relevant articles were censored in five domains: (1) bias arising from the randomization process; (2) bias due to deviations from intended interventions; (3) bias due to missing outcome data; (4) bias in the measurement of the outcome; (5) bias in the selection of the reported result. Disaccord would still be settled by communication. More details about the quality assessment of each included study can be found in the Appendix A.

### 2.5. Statistical Analysis

We generated network plots for different outcomes of different therapies in Stata (version 15.1). The Bayesian network meta-analysis was performed in R (version 4.1.3) software using the ‘rjags’ and ‘gemtc’ package. Using Markov chain Monte Carlo methods, four Markov chains were generated, and 200,000 iterations, with 50,000 burn-ins, as well as a thinning interval of 10, were used for each chain. Random effects were used to draw conservative conclusions [19,20]. Trace plots, density plots, and the Brooks–Gelman–Rubin method were used to test the convergence of the models [21]. Forest plots, probability diagrams, and the surface under the cumulative ranking curves (SUCRA) were used to explore the potentially optimal first-line treatment strategies for *EGFR*-mutated patients [22]. The SUCRA equals 1 if the treatment is certain to be the best, and 0 when it is certain to be the worst. The consistency assessment was unnecessary because there was no closed loop in our network. We also performed pairwise meta-analyses (PWMA) and checked the heterogeneity using the I^2^ test within the forest plots [23].

As different generations of TKIs have different efficacy and safety, we planned to develop the network meta-analysis separately in the first, second, and third generation TKI fields. If the number of studies in one of the fields was limited, PWMA would be performed in R (version 4.1.3) software using the ‘meta’ package.

## 3. Results

### 3.1. Study Selection and Characteristics

The initial retrieval yielded a total of 5622 studies, of which 5305 were eliminated due to duplication or abstract evaluation. After fully reviewing 317 pieces of literature, 24 studies, including a total of 4226 patients, finally met the selection criteria, most patients diagnosed with lung adenocarcinoma. (Figure 1). Notably, two studies were disregarded: one for its unconventional randomization, and the other for exploring the proposed sequencing of the identical treatment regimen [24,25].

A total of 21 studies considered the combination regimen compared with monotherapy of the first generation TKI (such as gefitinib, erlotinib, and icotinib). The networks were constructed regarding these studies. The treatment arms involved included TKI monotherapy (abbreviated as TKI), TKI plus pemetrexed (TKIplusP), TKI plus pemetrexed and carboplatin (TKIplusPC), TKI plus bevacizumab (TKIplusBev), TKI plus ramucirumab (TKIplusRam), TKI plus apatinib (TKIplusApa), TKI plus cryoablation (TKIplusCAb), TKI plus microwave ablation (TKIplusMAb), TKI plus linsitinib (TKIplusLin), TKI plus metformin (TKIplusMet), TKI plus olaparib (TKIplusOla), TKI plus radiation (TKIplusRt), and TKI plus radiotherapy and GM-CSF (TKIplusRtG). 

However, there were only three papers comparing the combination regimen to monotherapy of the second and third generations (two for 2nd generation TKI and one for 3rd generation TKI, respectively), making the construction of a network unfeasible. No studies involving multiple combination therapies were included. The main characteristics of all included studies are presented in Table 1. 

### 3.2. Network Meta-Analysis for PFS, OS, ORR and ≥Grade 3 AEs

A total of 10, 10, 13, and 9 treatment arms were included in the network meta-analysis for PFS, OS, ORR, and ≥Grade 3 AEs, respectively (Figure 2).

In terms of PFS (Figure 3A, Appendix A), TKI plus pemetrexed (HR = 0.63, 95% credible interval = 0.48–0.83)/pemetrexed and carboplatin (0.5, 0.4–0.61)/bevacizumab (0.6, 0.5–0.73)/ramucirumab (0.59, 0.42–0.82)/radiation (0.22, 0.13–0.38) achieved a significant benefit compared with TKI alone. TKI plus pemetrexed/pemetrexed and carboplatin/bevacizumab/ramucirumab did not differ in PFS between each other, but it is striking to note that the combination with radiation yielded a better benefit in PFS than any other treatment. However, the effects of the combination of linsitinib (1.38, 0.73–2.57) and metformin (1.04, 0.7–1.54) were similar to those of TKI monotherapy. Adding apatinib (0.71, 0.5–1.01)/olaparib (0.73, 0.49–1.07) was associated with a trend toward improvement, but additional exploratory trials are required to provide proof.

In terms of OS (Figure 3B, Appendix A), only the combination with pemetrexed and carboplatin (0.63, 0.43–0.86)/radiation (0.44, 0.23–0.83) was superior to TKI alone. There was no statistical significance across the combination treatments, except that TKI plus radiation showed better effects than TKI plus bevacizumab (0.48, 0.24–0.97)/metformin (0.38, 0.16–0.93)/olaparib (0.36, 0.15–0.88).

In terms of ORR (Figure 3C, Appendix A), despite the fact that the majority of combination therapies exhibited a trend toward improved ORR, only TKI plus microwave ablation (1.33, 1.01–1.82)/pemetrexed and carboplatin (1.23,1.1–1.39) achieved statistically significant results, possibly because of the good tumor regression effect that TKI monotherapy can provided. Notably, TKI combined with cryoablation, another ablation technique, also had a tendency to increase objective response rates (2.3, 0.9–5.54); however, survival data was lacking in both studies. Therefore, even though physical tumor destruction is more likely to result in tumor regression, it is still unknown if there is a survival benefit for patients; thus, additional research is required. 

In terms of ≥Grade 3 AEs (Figure 3D, Appendix A), all point estimates of the pooled RRs are greater than 1, indicating that all the combination therapies may increase the incidence of ≥Grade 3 AEs. The only statistically significant change was between TKI and TKI plus bevacizumab/apatinib. Notably, the pooled RRs of TKI plus pemetrexed/pemetrexed and carboplatin were rather high and close to the boundary of statistical significance (2.35, 0.96–6.05 and 2.11, 0.95–4.72, respectively). Each treatment exhibited unique AEs (Figure 4). TKI plus antiangiogenic drugs, such as bevacizumab, ramucirumab, and apatinib, caused more ≥Grade 3 hypertension, diarrhea, and proteinuria. TKI plus pemetrexed/pemetrexed and carboplatin results in more ≥Grade 3 anemia, neutropenia, and anorexia. The combination with apatinib was associated with a high risk of ≥Grade 3 hypertension (46.5%), and liver dysfunction was of frequent occurrence in patients with TKI plus linsitinib (37.2%).

### 3.3. Treatment Ranking for PFS, OS, ORR, and ≥Grade 3 AEs

The treatments were ranked according to SUCRA scores (Table 2). Regarding PFS, the three most effective treatments were TKI plus radiation, pemetrexed and carboplatin, and ramucirumab. The combinations of metformin/linsitinib were ranked behind TKI monotherapy, indicating inferior consequences. TKI plus radiation/pemetrexed and carboplatin remained the top two anticancer therapies for OS, demonstrating their gratifying antitumor efficacy. TKI cryoablation/microwave ablation/radiotherapy and GM-CSF showed the maximum effects on ORR, reflecting the benefit provided by local treatments. As expected, TKI alone was the medication with the lowest incidence of ≥Grade 3 AEs. Diagrams illustrating the probability of ranks for each treatment are available in Appendix A.

The comprehensive ranking plots using the SUCRA scores of PFS or OS with ≥Grade 3 AEs indicated that it might be difficult to find the “perfect” treatment because the top-right area of the plots was lacking in points, hinting that few treatments could achieve a high ranking in both efficacy and safety (Figure 5). Significantly, the combination with antiangiogenic drugs such as ramucirumab and bevacizumab may have the potential for wider clinical application due to its beneficial long-term antitumor effect and modest safety concerns.

### 3.4. Interruption Rate of TKI and Incidence of Developing T790M Mutation

In addition, the interruption rate of TKI owing to AEs was analyzed in the network (Appendix A). Compared to TKI monotherapy, the AEs of combination therapies did not lead to a significantly greater interruption rate of TKI. However, the combination of pemetrexed had the potential to interrupt the application of the TKI (RR = 2.2, 95% credible interval = 0.88–6.0).

The incidence of obtaining the T790M mutation in patients developing resistance to first generation TKIs was similar across several therapies. Nevertheless, the patients undergoing combination therapy had less tendency to develop the T790M mutation when experiencing the TKI resistance (Appendix A).

### 3.5. Network Meta-Analysis in Subgroup

Subgroup analyses of PFS and OS were also performed (Appendix A). All the subgroups (male/female, smoker/non-smoker, patients with brain metastases/patients without brain metastases, 19 deletion/21 Leu858Arg) were comparable in terms of optimal treatment selection. Regarding PFS and OS, pemetrexed and carboplatin were statistically the best addition to TKI, followed by ramucirumab and bevacizumab. This result conformed to that regarding the overall population, since research about TKI plus radiation lacks the information of these subgroups. Combination with bevacizumab, rather than pemetrexed and carboplatin, was more likely to be the optimal treatment in terms of OS in female patients, but this conclusion requires additional evidence.

### 3.6. Heterogeneity and Convergency Assessing

Heterogeneity was low (I^2^ < 50%) among most of RCTs included. However, high heterogeneity was detected between TKI versus TKI plus pemetrexed and carboplatin for OS (73.8%) and between TKI versus TKI plus bevacizumab for ≥Grade 3 AEs (69.3%) (Appendix A).

The model showed ideal convergence. The trace plots, density plots, and Brooks–Gelman–Rubin method diagram are presented in Appendix A.

### 3.7. Comparation of Treatments Based on the Second/Third Generation TKI

The details of the research is shown in Table 1. Only two studies compared afatinib plus cetuximab with afatinib monotherapy, and only one study compared osimertinib plus bevacizumab with osimertinib monotherapy. In consideration of the limited number of studies, network meta-analysis was not performed for afatinib and osimertinib. 

Compared to afatinib monotherapy, afatinib plus cetuximab showed superior ORR (pooled RR = 0.95, 95% confidence interval = 0.82–1.10) (Appendix A), but did not achieve significant benefits in either PFS (HR = 1.01, 95% confidence interval = 0.72–1.43) or OS (HR = 0.82, 95% confidence interval = 0.5–1.36) [50]. The addition of bevacizumab to osimertinib did not significantly improve PFS (HR = 0.86, 95% confidence interval = 0.53–1.40) [52].

## 4. Discussion

Combination therapy has emerged as a crucial method to overcome TKI resistance. However, there are few studies to compare the advantages and drawbacks of different combination schemes head-to-head. Some researchers have conducted meta-analyses of RCTs of EGFR-TKI combined with antiangiogenic agents and found that combination therapy significantly improved patients’ PFS, but was associated with higher AEs [53,54]. The combined benefit to patients was consequently insignificant. Other researchers conducted meta-analyses of first-line treatments for advanced lung cancer [10,11]. However, the findings are no longer applicable, due to the absence of results from several major RCTs. Our Bayesian network meta-analysis connected various treatment arms through the intermediary of TKI monotherapy, including the most recent RCTs.

Based on the first generation TKI, 12 combination therapies were enrolled. TKI plus pemetrexed and carboplatin/pemetrexed/bevacizumab/ramucirumab showed benefit in PFS or OS, whereas the combination with metformin/olaparib/linsitinib provided no advantages, but higher AEs. Although the addition of radiation had a significant effect on PFS and OS, it should be noted that only 133 patients from one RCT were included in the PFS and OS analyses. Its antitumor efficacy and safety require more evidence. In addition, combinations with other local treatments, such as cryoablation or microwave ablation, are worthy of an extensive study, as their clinical applicability is hindered by a paucity of studies. It is reasonable to assume that the combination of physical therapies may improve patients’ long-term prognosis. A combination with chemotherapy could bring definite benefits to ORR, PFS, and OS, but may also cause more ≥Grade 3 hematological AEs. Surprisingly, pemetrexed and carboplatin showed a stronger effect than pemetrexed alone, with no increase in ≥Grade3 AEs. Considering both efficacy and safety, our network meta-analysis revealed that combinations with macromolecular antiangiogenic medicines, such as bevacizumab or ramucirumab, may be more optimal. Apatinib also has the ability to inhibit angiogenesis [55], but its application as an addition to TKI should be further confirmed. The results of the subgroups were broadly consistent with those of the overall population. Utilization of osimertinib is associated with the development of T790M mutations in patients exhibiting resistance to first-generation TKIs. The incidence of T790M mutations in patients acquiring resistance to first generation TKIs was similar across several therapies. Nevertheless, the patients with combination therapy had less tendency to develop the T790M mutation when experiencing TKI resistance. However, most of RCTs achieved the data using the cell-free DNA test, and these had a small sample size. A retrospective study using formalin-fixed and paraffin-embedded tumor specimens suggested that the positive rate of T790M in the TKI plus bevacizumab group was significantly lower than that in the EGFR-TKI monotherapy group (51.5% vs. 35.5%, *p* = 0.0003) [56]. We can infer that the drug combination might influence future patient treatment decisions and clinical prognosis by altering the evolutionary trajectory of tumors.

Heterogeneity was low between most of our comparations, but was still detected in TKI versus TKI plus pemetrexed and carboplatin for OS. This may be attributed to the fact that OS could be greatly affected by the back-line treatment. There was considerable heterogeneity between TKI versus TKI plus bevacizumab for ≥Grade 3 AEs, possibly as a result of a study exclusively involving Westerners, which was distinct from prior studies comparing TKI versus TKI plus bevacizumab [41]. Considering the low heterogeneity between the great majority of studies and our use of a random effects model, the homogeneity and transitivity between treatment arms were acceptable. Our network lacks a closed loop; therefore, analyzing consistency is unnecessary.

Few studies explored the combination treatment based on the second or third generation for first line treatment. Both afatinib plus cetuximab and osimertinib plus bevacizumab failed to produce significant PFS or OS benefit compared with their corresponding TKI monotherapy. As for osimertinib, its combination approach has been validated in many studies of post-line therapy. A retrospective analysis revealed that the combination of osimertinib and chemotherapy improved the control of CNS lesions in individuals receiving backline treatment [57]. The ongoing phase III trial FLAURA2 compares first-line osimertinib plus platinum-based chemotherapy with osimertinib alone in *EGFR*-mutated NSCLC. The first results published demonstrated the safety and tolerability of this combination [58]. There is evidence from preclinical studies that VEGF/VEGF receptor inhibitors can boost the effectiveness of EGFR TKIs [59]. Second-line treatment of T790M patients with bevacizumab in combination with osimertinib did not result in a prolonged progression-free survival (PFS) compared to osimertinib alone [60]; Nonetheless, other clinical trials of first-line treatment are ongoing (NCT02803203). In the TATTON trial, the combination of osimertinib and anti-PD-L1 durvalumab was associated with a significant number of immune-mediated adverse events, especially in interstitial lung disease (ILD) (NCT02143466) [61]. In the future, it is anticipated that more combination therapies based on osimertinib will be proposed.

Our meta-analysis has several limitations. Firstly, we regarded gefitinib, erlotinib, and icotinib as the same treatment arms, which may introduce bias. However, several studies have revealed that three first generation TKIs had similar efficacy and safety. The previously published network meta-analysis likewise found no statistically significant difference between gefitinib, erlotinib, and icotinib in PFS, OS, ORR, and ≥Grade 3 AEs [10]. Secondly, we did not include 2nd and 3rd generation TKI, since they are irreversible inhibitors that are fundamentally distinct from first generation drugs and hence, cannot be combined in a network. Thirdly, in order to cover as many treatment regimens as possible, certain studies with small sample sizes and insufficient outcome reports were included. Therefore, caution should be exercised when interpreting the pooled HR or RR near the statistical boundary.

## 5. Conclusions

In our network meta-analysis, TKI combination of radiation/pemetrexed and carboplatin could provide the best antitumor effects among the first generation TKI-based first-line treatments. Taking safety into account, ramucirumab and bevacizumab may be the ideal additions to the first generation TKI. Combination therapies based on the second or third generation of TKIs require further investigation.

## Figures and Tables

**Figure 1 cancers-14-04894-f001:**
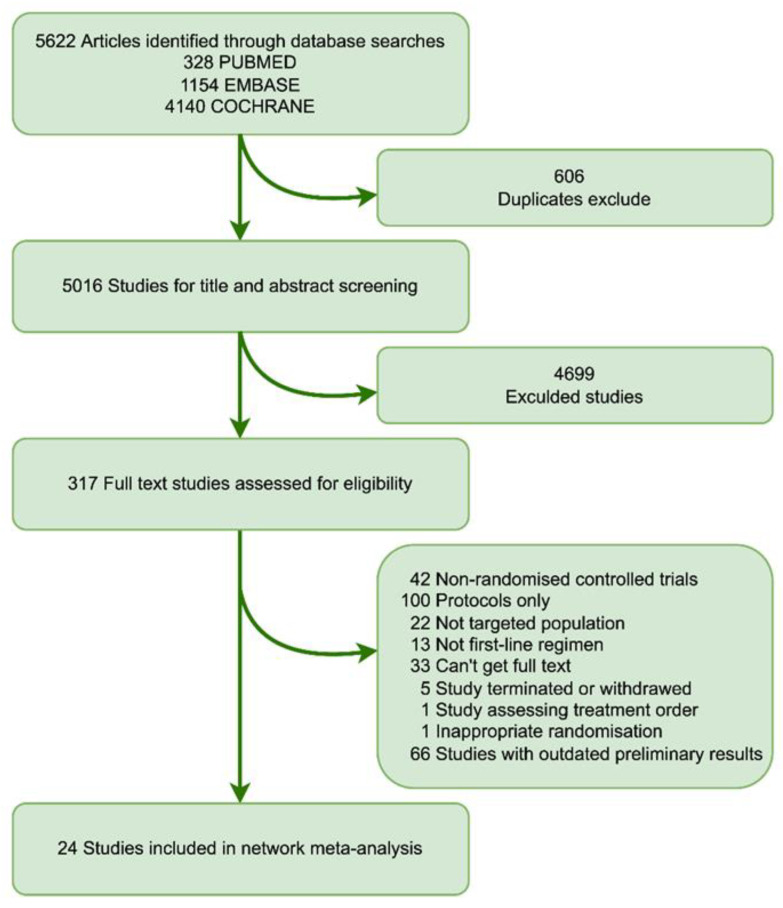
Flow chart depicting the process used for study selection.

**Figure 2 cancers-14-04894-f002:**
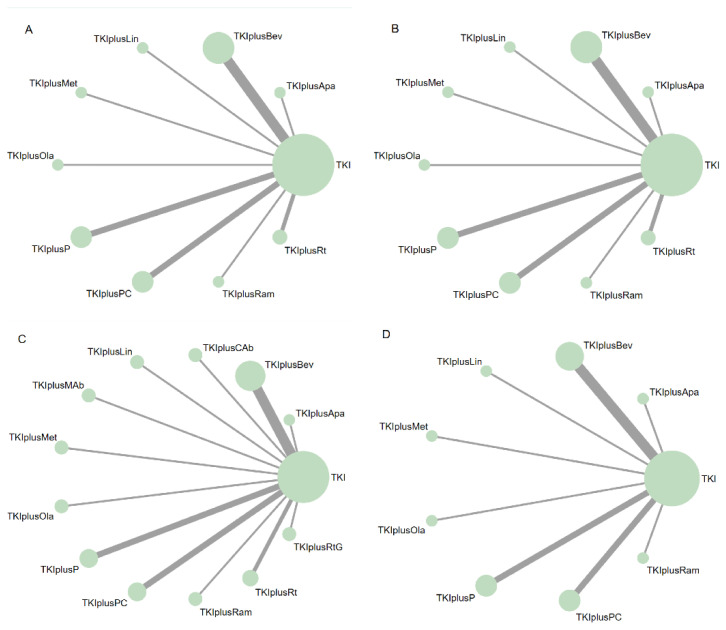
Network diagrams of comparisons of different outcomes of treatments in different groups of patients with advanced epidermal growth factor receptor (*EGFR*) mutated non-small cell lung cancer (NSCLC). (**A**) Comparisons of progression free survival (PFS) in patients with advanced *EGFR* mutated NSCLC. (**B**) Comparisons of overall survival (OS) in patients with advanced *EGFR* mutated NSCLC. (**C**) Comparisons of objective response rate (ORR) and adverse events in patients with advanced *EGFR* mutated NSCLC. (**D**) Comparisons of adverse events of grade 3 or higher (≥Grade 3 AEs) in patients with advanced *EGFR* mutated NSCLC. The node size is proportional to the total number of patients receiving treatment. Each line represents a type of head-to-head comparison. The width of lines is proportional to the number of trials comparing the connected therapies. TKI: tyrosine kinase inhibitors, representing first-generation EGFR-TKIs in this network meta-analysis (including gefitinib, erlotinib, and icotinib). TKIplusP: TKI plus pemetrexed; TKIplusPC: TKI plus pemetrexed and carboplatin; TKIplusBev: TKI plus bevacizumab; TKIplusRam: TKI plus ramucirumab; TKIplusApa: TKI plus apatinib; TKIplusCAb: TKI plus cryoablation; TKIplusMAb: TKI plus microwave ablation; TKIplusLin: TKI plus linsitinib; TKIplusMet: TKI plus metformin; TKIplusOla: TKI plus olaparib; TKIplusRt: TKI plus radiation; TKIplusRtG: TKI plus radiotherapy and GM-CSF.

**Figure 3 cancers-14-04894-f003:**
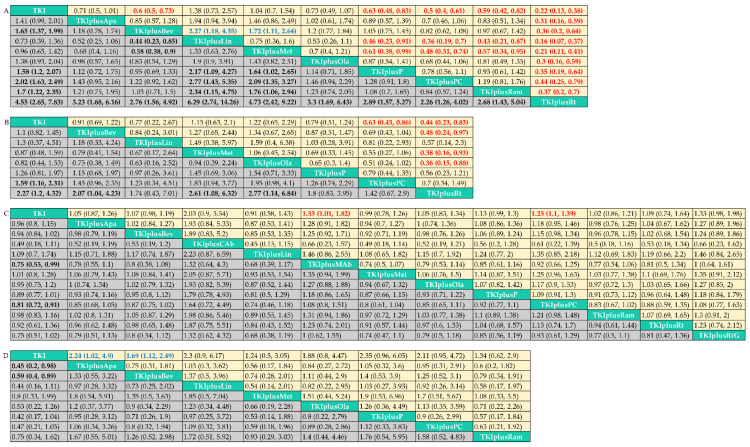
Pooled estimates of the network meta-analysis. Data in each cell are hazard or risk ratios (95% credible intervals) for the comparison of column-defining treatment versus row-defining treatment. Hazard ratios less than 1 and risk ratios of objective response rate greater than 1 favor the column-defining treatment, while risk ratios of adverse events of grade 3 or higher that are less than 1 favor the column-defining treatment. Significant results are in bold red (in favor of the column-defining treatment) or blue (opposed to the column-defining treatment). (**A**) Progression free survival in all patients included in the analysis; (**B**) Overall survival in all patients included in the analysis; (**C**) Objective response rate in all patients included in the analysis; (**D**) Adverse events of grade 3 or higher rate in all patients included in the analysis. TKI: tyrosine kinase inhibitors, representing first-generation EGFR-TKIs in this network meta-analysis (including gefitinib, erlotinib, and icotinib). TKIplusP: TKI plus pemetrexed; TKIplusPC: TKI plus pemetrexed and carboplatin; TKIplusBev: TKI plus bevacizumab; TKIplusRam: TKI plus ramucirumab; TKIplusApa: TKI plus apatinib; TKIplusCAb: TKI plus cryoablation; TKIplusMAb: TKI plus microwave ablation; TKIplusLin: TKI plus linsitinib; TKIplusMet: TKI plus metformin; TKIplusOla: TKI plus olaparib; TKIplusRt: TKI plus radiation; TKIplusRtG: TKI plus radiotherapy and GM-CSF.

**Figure 4 cancers-14-04894-f004:**
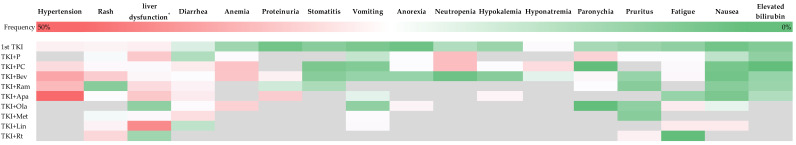
The frequency of adverse events of grade 3 or higher (≥Grade 3 AEs) profile in relation to the incidence (%) of each specific adverse event based on the population of each treatment we included. Grey color block means not available. * When not reported, liver dysfunction is manifested by an increase in glutamic pyruvic transaminase or glutamic oxaloacetic transaminase, as it was reported in most studies. 1stTKI: tyrosine kinase inhibitors, representing first-generation EGFR-TKIs in this network meta-analysis (including gefitinib, erlotinib, and icotinib); TKIplusP: TKI plus pemetrexed; TKIplusPC: TKI plus pemetrexed and carboplatin; TKIplusBev: TKI plus bevacizumab; TKIplusRam: TKI plus ramucirumab; TKIplusApa: TKI plus apatinib; TKIplusOla: TKI plus olaparib; TKIplusMet: TKI plus metformin; TKIplusLin: TKI plus linsitinib; TKIplusRt: TKI plus radiation.

**Figure 5 cancers-14-04894-f005:**
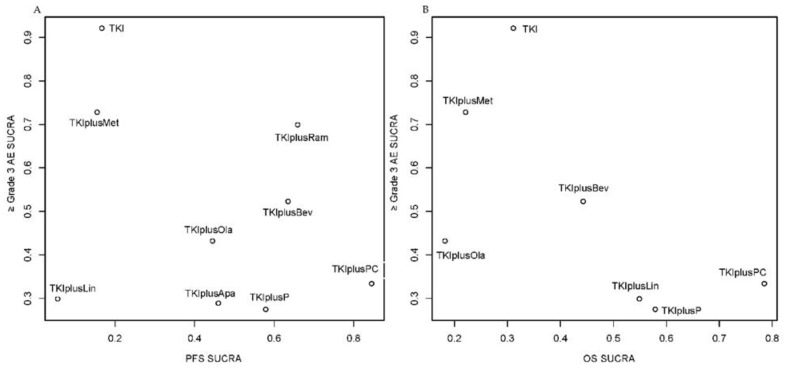
Ranking plot based simultaneously on efficacy (x-axis: SUCRA value of progression free survival (PFS) or overall survival (OS)) and tolerability (y-axis: SUCRA value of adverse events of grade 3 or higher (≥Grade 3 AEs)). (**A**) Ranking plot based simultaneously SUCRA value of PFS and SUCRA value of ≥Grade 3 AE; (**B**) Ranking plot based simultaneously SUCRA value of OS and SUCRA value of ≥Grade 3 AE. SUCRA, surface under the cumulative ranking curves; TKIplusP: TKI plus pemetrexed; TKIplusPC: TKI plus pemetrexed and carboplatin; TKIplusBev: TKI plus bevacizumab; TKIplusRam: TKI plus ramucirumab; TKIplusApa: TKI plus apatinib; TKIplusLin: TKI plus linsitinib; TKIplusMet: TKI plus metformin; TKIplusOla: TKI plus Olaparib.

**Table 1 cancers-14-04894-t001:** Baseline characteristics of studies included in the network meta-analysis of patients with advanced epidermal growth factor receptor (*EGFR*) mutated non-small cell lung cancer.

Study	Phase	Region	Sample Size (No.)	Intervention Arm	Control Arm	*EGFR* Mutation	
						19del	L858R
Y. Cheng et al. 2016 [26,27]	II	East Asia	126/65	Pemetrexed 500 mg/m^2^ every 3 weeks + gefitinib 250 mg once a day	Gefitinib 250 mg once a day	65/40	52/23
C. An et al. 2016 [28]	II	China	45/45	Pemetrexed 500 mg/m^2^ every 3 weeks + gefitinib 250 mg once a day	Placebo 500 mg/m^2^ every 3 weeks + gefitinib 250 mg once a day	16/17	29/28
B. Han et al. 2017 [29,30]	II	China	40/41	Pemetrexed 500 mg/m^2^ + carboplatin (AUC 5) every 3 weeks + gefitinib 250 mg/day once a day	Gefitinib 250 mg once a day	21/21	19/20
L. Xu et al. 2019 [31]	II	China	90/89	Pemetrexed 500 mg/m^2^ + carboplatin (AUC 5) every 3 weeks + icotinib 125 mg, three times a day	Icotinib 125 mg, three times a day	51/52	38/37
Y. Hosomi et al. 2019 [32]	III	Japan	169/172	Pemetrexed 500 mg/m^2^ + carboplatin (AUC 5) every 3 weeks + gefitinib 250 mg/day once a day	Gefitinib 250 mg once a day	93/95	69/67
V. Noronha et al. 2019 [33]	III	India	174/176	Pemetrexed 500 mg/m^2^ + carboplatin (AUC 5) every 3 weeks + gefitinib 250 mg/day once a day	Gefitinib 250 mg once a day	107/109	60/60
T. Seto et al. 2014 [34,35,36]	II	Japan	75/77	Erlotinib 150 mg once a day + bevacizumab 15 mg/kg every 3 week	Erlotinib 150 mg once a day	40/40	35/37
H. Saito et al. 2019 [37,38]	III	Japan	112/112	Erlotinib 150 mg once a day + bevacizumab 15 mg/kg every 3 week	Erlotinib 150 mg once a day	56/55	56/57
T. E. Stinchcombe et al. 2019 [36]	II	America	43/45	Erlotinib 150 mg once a day + bevacizumab 15 mg/kg every 3 week	Erlotinib 150 mg once a day	29/30	14/15
Q. Zhou et al. 2021 [39]	III	China	157/154	Erlotinib 150 mg once a day + bevacizumab 15 mg/kg every 3 week	Erlotinib 150 mg once a day	82/79	75/75
K. Nakagawa et al. 2019 [40]	III	Global	224/225	Erlotinib 150 mg once a day + ramucirumab 10 mg/kg every 2 week	Erlotinib 150 mg once a day + placebo 10 mg/kg every 2 week	123/120	99/105
M. C. Piccirillo et al. 2022 [41]	III	Italy	80/80	Erlotinib 150 mg once a day + bevacizumab 15 mg/kg every 3 week	Erlotinib 150 mg once a day	44/44	34/32
X. Gu et al. 2011 [42]	II	China	18/18	Gefitinib 250 mg once a day + cryoablation	Gefitinib 250 mg once a day	NA	NA
B. Yu et.al. 2019 [43]	II	China	55/55	Gefitinib 250 mg once a day + microwave ablation	Gefitinib 250 mg once a day	NA	NA
N. B. Leighl et al. 2017 [44]	II	Global	44/44	Erlotinib 150 mg once a day + linsitinib 150 mg twice a day	Placebo 150 mg twice daily plus erlotinib 150 mg once daily	26/25	18/19
L. Li et al. 2019 [45]	II	China	112/111	Gefitinib 250 mg once a day + metformin was 500 mg after meal daily	Gefitinib 250 mg once a day	54/61	53/43
R. G. Campelo et al. 2020 [46]	II	Spain and Mexico	91/91	Gefitinib 250 mg once a day + olaparib 200 mg every 28-day	Gefitinib 250 mg once a day	57/52	25/35
X. Zheng et al. 2016 [47]	II	China	38/38	Erlotinib 150 mg once a day or icotinib 125 mg three times a day + 3D conformal radiation	Erlotinib 150 mg once a day or Icotinib 125 mg three times a day	22/21	18/17
X. Wang et al. 2022 [48]	III	China	68/65	Gefitinib 250 mg once a day or erlotinib 150 mg once a day or icotinib 125 mg three times a day + radiation	Gefitinib 250 mg once a day or erlotinib 150 mg once a day or icotinib 125 mg three times a day	45/47	23/18
Y. Qiu et al. 2020 [49]	II	China	21/21	Icotinib 125 mg three times a day + radiotherapy and GM-CSF	Icotinib 125 mg three times a day	15/13	6/8
S. B. Goldberg et al. 2020 [50]	II	America	83/85	Afatinib 40 mg once a day + cetuximab 500 mg/m^2^ every 2 weeks	Afatinib 40 mg once a day	53/54	30/31
A. B. Cortot et al. 2021 [51]	II	France	58/59	Afatinib 40 mg once a day + cetuximab 500 mg/m^2^ every 2 weeks	Afatinib 40 mg once a day	32/33	24/23
H. Kenmotsu et al. 2022 [52]	II	Japan	61/61	Osimertinib 80 mg once a day + bevacizumab 15 mg/kg every 3 week	Osimertinib 80 mg once a day	35/36	26/25

Data are expressed as intervention/control, unless otherwise indicated. NA: not available; GM-CSF: granulocyte macrophage colony-stimulating factor.

**Table 2 cancers-14-04894-t002:** Ranking of TKI and several combination treatments.

PFS		OS		ORR		≥Grade 3 AE	
Treatment	SUCRA	Treatment	SUCRA	Treatment	SUCRA	Treatment	SUCRA
TKIplusRt	0.999	TKIplusRt	0.930	TKIplusCAb	0.912	TKI	0.921
TKIplusPC	0.845	TKIplusPC	0.785	TKIplusMAb	0.799	TKIplusMet	0.728
TKIplusRam	0.659	TKIplusP	0.579	TKIplusRtG	0.784	TKIplusRam	0.699
TKIplusBev	0.635	TKIplusLin	0.549	TKIplusPC	0.746	TKIplusBev	0.523
TKIplusP	0.579	TKIplusBev	0.443	TKIplusP	0.569	TKIplusOla	0.432
TKIplusApa	0.460	TKI	0.311	TKIplusRt	0.461	TKIplusPC	0.334
TKIplusOla	0.445	TKIplusMet	0.221	TKIplusBev	0.439	TKIplusLin	0.299
TKI	0.167	TKIplusOla	0.182	TKIplusOla	0.391	TKIplusApa	0.289
TKIplusMet	0.155	TKIplusApa	NA	TKIplusApa	0.376	TKIplusP	0.275
TKIplusLin	0.056	TKIplusRam	NA	TKIplusRam	0.315	TKIplusCAb	NA
TKIplusCAb	NA	TKIplusCAb	NA	TKIplusMet	0.262	TKIplusMAb	NA
TKIplusMAb	NA	TKIplusMAb	NA	TKI	0.228	TKIplusRtG	NA
TKIplusRtG	NA	TKIplusRtG	NA	TKIplusLin	0.219	TKIplusRt	NA

SUCRA: surface under the cumulative ranking curve; it equals 1 when a treatment is certain to be the best and 0 when a treatment is certain to be the worst. NA means not available.

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
