# Peer review of "Efficacy and Safety of Epidermal Growth Factor Receptor (EGFR)-Tyrosine Kinase Inhibitor Combination Therapy as First-Line Treatment for Patients with Advanced EGFR-Mutated, Non-Small Cell Lung Cancer: A Systematic Review and Bayesian Network Meta-Analysis"

_cancers, 2022, doi:10.3390/cancers14194894_

Round 1
Reviewer 1 Report
The authors have presented a very comprehensive review and analysis of combination therapy for non small cell lung cancer. The methodology and statistical analysis is well done and well explained.
I would like the authors to reword some of the discussion where they continue to put forward hypotheses when they are better served with inferences or explanations. For eg on Line 375 the authors hypothesize about future patient treatment. Similarly on line 358.
Also, a discussion of similar meta analyses would be helpful to put the current study in it's proper context.
Lastly, the title would benefit from being shortened
Reviewer 2 Report
In present review, authors have compared the efficacy of different treatments in combination with tyrosine kinase inhibitors in NSCLC. I have several reservations, my comments are appended as below:
1. Simple summary- note the type of pathology being studied.
2. Authors appears to compare the TKI with radiation and chemotherapy. Immunotherapy is also routinely followed in NSCLC, authors should also include this.
3. Does any report notes additional cofounders such as BMI, age, sex, ethnicity of patients?
4. Does any study includes recurrent tumors?
5. How PFS and OS was defined?
6. My another major concern making this study potentially unfit for publication is similar reports exists in literature, for instance- PMID: 32714857, PMID: 34123814
Reviewer 3 Report
Many studies have been done about randomized controlled trials of Epidermal Growth Factor Receptor-Tyrosine Kinase Inhibitor (EGFR-TKI) in combination with other treatment have been conducted to compare the benefits of efficacy and safety of combination therapy over EGFR-TKI monotherapy. In this study, the authors performed network meta-analysis that is widely used in the absence of data from head-to-head trials, integrating the most recent results from RCTs and synthesizing indirect evidence to investigate the efficacy and safety of combination therapy of EGFR-TKI as first-line treatment against advanced EGFR mutated Non-small cell lung carcinoma (NSCLC). As a result, the authors revealed that radiation combined with EGFR-TKI is superior to other combination treatments in progression free survival (PFS) and overall survival (OS) and TKI combined with chemotherapy may improve objective response rate (ORR), PFS, and OS but may increase grade 3 or higher hematological side events. There is a limitation of this study is that certain data of new generation of TKI is not included and using data with small sample sizes and insufficient outcome reports were included. However these analysis, such as tables and figures, and statistical data showed the significant implications for the clinical management of patients with NSCLC, providing insights and evidences for medical decision.
Round 2
Reviewer 2 Report
accept